# Potential Transcriptional Enhancers in Coronaviruses: From Infectious Bronchitis Virus to SARS-CoV-2

**DOI:** 10.3390/ijms25158012

**Published:** 2024-07-23

**Authors:** Roberto Patarca, William A. Haseltine

**Affiliations:** 1ACCESS Health International, 384 West Lane, Ridgefield, CT 06877, USA; william.haseltine@accessh.org; 2Feinstein Institutes for Medical Research, 350 Community Dr, Manhasset, NY 11030, USA

**Keywords:** coronavirus, enhancer, long-range RNA interactions, IBV, SARS-CoV-2, viral attenuation, host immune evasion

## Abstract

Coronaviruses constitute a global threat to human and animal health. It is essential to investigate the long-distance RNA-RNA interactions that approximate remote regulatory elements in strategies, including genome circularization, discontinuous transcription, and transcriptional enhancers, aimed at the rapid replication of their large genomes, pathogenicity, and immune evasion. Based on the primary sequences and modeled RNA-RNA interactions of two experimentally defined coronaviral enhancers, we detected via an in silico primary and secondary structural analysis potential enhancers in various coronaviruses, from the phylogenetically ancient avian infectious bronchitis virus (IBV) to the recently emerged SARS-CoV-2. These potential enhancers possess a core duplex-forming region that could transition between closed and open states, as molecular switches directed by viral or host factors. The duplex open state would pair with remote sequences in the viral genome and modulate the expression of downstream crucial genes involved in viral replication and host immune evasion. Consistently, variations in the predicted IBV enhancer region or its distant targets coincide with cases of viral attenuation, possibly driven by decreased open reading frame (ORF)3a immune evasion protein expression. If validated experimentally, the annotated enhancer sequences could inform structural prediction tools and antiviral interventions.

## 1. Introduction

Coronaviruses employ complex replicative strategies involving long-range RNA-RNA interactions. These strategies, which include genome circularization, discontinuous transcription, and viral enhancers, approximate regulatory sequences in their large genomes influencing replication, pathogenicity, and immune evasion [1,2,3,4,5,6,7].

Genome circularization approximates regulatory sequences in the 5′- and 3′-untranslated regions (UTRs), via the interaction of complementary sequences in the UTRs facilitated by viral and host protein bridges during the synthesis of subgenomic negative-sense strands [5,6,7,8,9,10,11,12,13,14,15,16,17,18,19,20,21,22]. Discontinuous transcription generates a nested set of subgenomic RNAs via transcription regulatory sequence (TRS)-dependent template switching [7]. The 5′-untranslated region (UTR) TRS leader interacts with homologous TRS body elements upstream of structural and accessory genes in the last third of the genome, driven by the extent of the TRSs’ base-pairing, viral and host protein–RNA binding, and high-order RNA-RNA interactions [6,7,8,23,24,25].

Regarding viral enhancer elements, a long-range interaction spanning approximately 26 kilobases was identified in the *Alphacoronavirus* transmissible gastroenteritis virus (TGEV) that was required for efficient transcription of its subgenomic mRNA encoding the nucleocapsid *(N*) gene, its most abundant subgenomic mRNA, despite its less robust TRS-L-TRS-B interaction (Figure 1) [24,25]. This nonanucleotide TGEV enhancer is conserved among other *Alphacoronaviruses*, such as feline infectious peritonitis virus and canine enteric coronavirus, but not porcine epidemic diarrhea virus (PEDV) or human coronavirus (hCoV)-229E [25]. The TGEV enhancer, via long-distance RNA-RNA interactions between complementary proximal and distal elements, brings TRS-L and TRS-B upstream of the *N* gene closer for discontinuous transcription, resulting in higher *N* gene expression levels [24,25]. The coronaviral nucleocapsid (N) protein, in turn, has been postulated to act as a transactivator of gene expression similar to the lentiviral Tat protein, via interactions with the 5′-UTR leader sequence and with viral and host proteins in the replication–transcription complex [26].

A nonanucleotide enhancer element (UUUAUAAAC) was also characterized in MHV, a *Betacoronavirus* (subgenus *Embecovirus*) just distal to its TRS-L [27]. However, the full mechanism underlying the enhancer activity has not been characterized.

The high prevalence of long-range RNA-RNA interactions in both the coding and non-coding regions of coronaviruses and other RNA viruses, including insect nodaviruses [28] and plant tombusviruses [29,30], the phylogenetic conservation of the involved RNA motifs, and the considerable level of global organization [31] in replicative strategies in coronaviruses reinforces the relevance of these high-order structures for the viral transcription of numerous distinct subgenomic mRNAs.

Long-distance RNA-RNA interactions contribute to RNA’s three-dimensional structure complexity and gene expression regulation, translation, and resistance to degradation, among other functions [32]. Although genetic and biochemical analyses have confirmed the functional importance of many of these structures, their precise roles remain to be fully defined [33,34,35]. Understanding the noncoding functions of viral RNA during viral replication provides new insights into virus–host interactions and therapeutic targets, which is the focus of this study, predicting coronaviral enhancers and analyzing, in the case of IBV, their impact using published cases of viral attenuation or its reversal.

## 2. Results

### 2.1. TGEV Enhancer-Based Model for Murine Hepatitis Virus (MHV) Enhancer and Potentially Bovine Coronavirus

We used the TGEV enhancer model to ascertain if the MHV enhancer could use a similar mechanism to that of TGEV in Figure 1. We searched for sequences similar to the MHV enhancer immediately distal to the 5′-leader in the MHV genome and found one in the region encoding the nonstructural protein (NSP)14 in *ORF1ab* (Figure 2A). Therefore, the characterized 5′-UTR MHV enhancer element [27] could pair a complementary sequence in the region encoding NSP14 at the end of *ORF1ab* with a minimum free energy (ΔG) of −6.9 kcal/mol. This pairing would bring closer together the TRS-L and the TRS-B preceding the *ORF2* gene, encoding a phosphodiesterase that antagonizes host interferon (IFN) signaling via antagonism of the 2′,5′-oligoadenylate synthetase (OAS)–RNase L pathway [35,36,37]. NS2 is a critical determinant of MHV strain A59 liver tropism in C57BL/6 (B6) mice and is required for the virus to cause hepatitis [36,38].

Another *Embecovirus*, the bovine coronavirus, has a matching potential enhancer element to that of MHV, also immediately distal to the leader sequence (Figure 2B). As in MHV, said enhancer element could pair with more distal genomic sequences. Two distal sequences are in the *NSP3* gene, and a third is in *NSP12*, encoding the RNA-dependent RNA polymerase. The pairings involving the first *NSP3* sequence to the second more distal *NSP3* one and that in *NSP12* are more stable, i.e., have a lower minimum free energy, than those involving the 5′-UTR sequence and would bring the TRS-L closer to the TRS-B preceding the segments encoding from the NSP13 helicase, NSP14 exoribonuclease, and NSP15 endoribonuclease to the NSP16 2′-O-methyltransferase, all involved in RNA genome replication and immune evasion and beyond to accessory and structural genes.

The proposed MHV and bovine coronavirus enhancer model is reminiscent of that described in the tomato bushy stunt virus, a positive-strand RNA tombusvirus [39]. In this tombusvirus, a 12-nucleotide sequence located ~1000 nucleotides upstream from the initiation site of subgenomic mRNA2 synthesis is required for the accumulation of said RNA through a long-distance RNA-RNA base-pairing interaction with a sequence located just 5′ to the transcription initiation site [40].

Eight of the nine nucleotides in the documented MHV and potential bovine coronavirus enhancer elements comprise a sequence that reads identically in the sense and antisense directions, with complementary halves (Figure 2C).

### 2.2. Potential Enhancer in Infectious Bronchitis Virus (IBV)

We determined if other coronaviruses have a sequence similar in primary and secondary structure to the MHV enhancer (Figure 2C), reading the same in the sense and antisense strands with complementary halves. We found this sequence in the region encoding NSP16 of IBV (Figure 3A), a *Gammacoronavirus*, the most ancient genus among coronaviruses [39]. NSP16 and NSP7 block antigen presentation, inhibiting adaptive immunity, and NSP16, together with NSP13 and NSP14, are involved as a 2′-O-methyltransferase in capping viral mRNA, which limits its recognition by the innate immune system [41]. The duplex-forming sequence reading the same in the sense and antisense direction with complementary halves that can form a duplex (ΔG = −7.4 kcal/mol) is part of an extended duplex-forming region (ΔG = −13.3 kcal/mol) (Figure 3B). Two similar subsequences (expect = 0.2) that could pair with the *NSP16* duplex are found proximal to it in the region encoding the first protein of ORF1a and distal to it in that encoding the spike (S) protein of IBV and preceding the *ORF3a* gene, which encodes an accessory protein involved in host immune evasion, specifically interferon resistance [42,43] (Figure 3C).

Because enhancers in viruses and eukaryotes show variation with functional consequences [44], we analyzed for the presence of mutations in the extended *NSP16* duplex among avian IBVs in the GenBank database and their effect on duplex minimum free energy and the encoded amino acid sequence as potential attenuation determinants (Figure 4; GenBank accession numbers, geographical locations, and pathogenic characteristics are provided in Appendix A). Some mutations did not affect the duplex minimum free energy, while others decreased or increased it. Attenuation is unlikely to be driven by amino acid sequence changes, because 91.8% of IBV strains in GenBank have a conserved amino acid sequence in the segment encoded by the *NSP16* extended duplex at the 5′ end of the reading frame (Figure 4).

#### 2.2.1. Variation in *NSP16* Duplex, Its Distal Binding Sequence in S, or Both, Consistent with Reported Cases of Viral Attenuation or Its Reversal

Using published cases of viral attenuation or its reversal, we analyzed whether changes in the *NSP16* duplex, the distal binding sequence in *S*, or both, would be consistent with the phenotypes described. Figure 5 provides examples of the three scenarios of such changes consistent with attenuation or its reversion documented in the literature. These findings may pinpoint the *NSP16* duplex as another potential pathogenicity determinant underlying attenuation or conversion to a pathogenic phenotype. However, these changes occurred in the context of others suggested to be associated with attenuation or its reversal.

In the first example of the attenuation of a pathogenic strain consistent with mutations in the *NSP16* duplex (Figure 5), after serial passages in chicken embryos, the Mass41 attenuated vaccine strain (Mildvac-H) was derived from the Massachusetts strain Mass/Mass41/41 wild-type strain [45]. The Mass41 attenuated vaccine strain has two mutations in the extended *NSP16* duplex (duplex group 36, ΔG of −7.4 Kcal/mol, GenBank accession number GQ504725.1, Figure 4) relative to the wild-type strain (duplex group 1, ΔG of −13.3 Kcal/mol, GenBank accession number GQ504724.1). The sequence to which the open *NSP16* duplex could bind in the spike (*S*) gene is conserved in both strains, resulting in a less favorable minimum free energy of the pairing for the attenuated vaccine strain (ΔG = −3.8 Kcal/mol) than the wild-type strain (ΔG = −7.4 Kcal/mol). The expression of ORF3a is predicted to be less or not enhanced in the attenuated strain, diminishing its interferon resistance-mediated host immune evasion ability [43]. The amino acid sequence encoded by the extended duplex segment does not differ between strains (Figure 4), underscoring the nucleotide changes as potential pathogenic determinants.

Another publication provides the second example in Figure 5, namely, of the re-emergence of pathogenicity consistent with mutations in the *NSP16* duplex and the potential binding site in spike (*S*) in an attenuated vaccine strain. IBV 3575/08 (GenBank accession number KX266757), with strong respiratory and renal pathogenicity, was isolated from chicken broilers vaccinated with the attenuated viral vaccine derived from a Taiwan strain 2575/98 (GenBank accession number MN128087.1), with which it shares a high similarity in structural proteins such as the spike; however, the amino acid differences confer distinct antigenicity and low cross-protection [46]. Nonstructural proteins involved in host immune evasion, such as those encoded by the ORFs *3a*, *3b*, and *5* genes, are not highly similar between strains. In terms of the extended *NSP16* duplex, IBV 3575/08 has three mutations (duplex ΔG of −6.6 Kcal/mol; group 16 in Figure 4), while the attenuated parent strain 2575/98 has only one mutation (duplex ΔG = −12.5 Kcal/mol; group 22 in Figure 4). However, the free minimum energy (ΔG) of the interaction between the open *NSP16* duplex and *S* gene sequence is more favorable for 3575/08 (ΔG = −9.8 Kcal/mol) than for the parent strain 2575/98 (ΔG = −2.0 Kcal/mol) (Figure 5). The predicted lower enhancer effect for expression of the *ORF 3a* and *3b* genes in the 2575/98 strain would be consistent with the observation that chickens infected with 3575/08 have a delayed expression of a set of the host’s innate immune genes, i.e., better host immune evasion by the virus. In contrast, the expression of host innate immune genes is quicker and more efficient in chickens infected with 2575/98, reflecting worse host immune evasion by the virus, possibly secondary to a lower or absent enhancer effect by the open *NSP16* duplex [46]. An alternative or additional explanation is that the decreased immune evasion capability of the 2575/98 strain relative to 3575/08 could be due to differences in the ORF 3a, 3b, and 5 protein sequences. Amino acid sequences in said nonstructural or other genes may also underlie the attenuation of the vaccine strain 2575/98 relative to the wild-type strain 2575/98 (GenBank accession number DQ646405.2), because both have the same *NSP16* extended duplex sequence (both in group 22, Figure 4) and a possible binding sequence in S. All three strains have the same NSP16 amino acid sequence in the segment encoded by the extended duplex (Figure 4, two are in group 16 and one in group 22). Therefore, the differences in pathogenicity are not related to variations in the NSP16 amino acid sequence in the region encoded by the extended duplex.

The third example in Figure 5 involves the attenuation of a pathogenic strain consistent with mutations in the potential binding site in *S*. IBV strain E160_YN (GenBank accession number MK644086.1) is a strain with an *NSP16* duplex corresponding to group 1 (ΔG = −13.3 Kcal/mol) but with mutations in the potential binding site in *S* that could reduce or obliterate the enhancement of *ORF 3a* expression (a low ΔG for pairing between the *NSP16* core duplex and the binding sequence in *S*). However, the *S* gene has a premature termination codon, and the virus does not encode ORF 5a because of an 81-nucleotide deletion [47,48], both suggested to contribute to the attenuation.

Other published cases of viral attenuation do not involve changes in the IBV *NSP16* extended duplex, such as the attenuated commercial vaccine strain TW2575/98vac derived from the wild strain TW2575/98w (both duplex group 1, Figure 4, Appendix A) [49]; the attenuated Ark DPI 101 derived from the virulent Ark DPI 11 (both duplex group 36, Figure 4, Appendix A) [50]; the less pathogenic Mass variant 15SK-02 relative to the more pathogenic 15AB-01 (both duplex group 1, Figure 4 and Appendix A) [51]; and the attenuated CK/CH/LDL/97I P115 strain derived from the CK/CH/LDL/97I P5 strain (both duplex group 36, with the same binding sequences) [52]. These observations underscore the existence of multiple mechanisms for viral attenuation.

#### 2.2.2. Further Analysis of *NSP16* Duplex Variation and Its Possible Association with Viral Attenuation or Its Reversal

To further analyze the possible association between an extended *NSP16* duplex variation and viral attenuation, we assessed the distribution of numbers of IBV strains in the GenBank database according to the ΔG of their extended *NSP16* duplex (Figure 6).

Overall, there is a dichotomization of the distribution of the number of GenBank entries into two clusters around the most prevalent values of minimum free energy, corresponding to groups 1 (reference) and 36 in Figure 4 (Figure 6 top panel). The temporal distributions of the numbers of groups 1 and 36 entries were similar (Figure 6 middle panel), with an exponential correlation favoring greater numbers of IBV strains with a higher minimum free energy over time (Figure 6 bottom panel). This correlation is consistent with vaccination campaigns with attenuated strains that can spread among poultry and the emergence of novel pathogenic strains via mutations and recombination within and among wild-type and vaccine strains, including attenuated and revertant [46,47,52,53,54,55,56,57,58,59,60,61,62,63,64,65,66,67,68,69,70,71,72,73,74,75,76,77,78,79,80,81,82,83]. Some of these mutation and recombination events may allow IBV to evolve to infect other species, as demonstrated with primate cells in vitro [84].

The possible relationship between an extended *NSP16* duplex ΔG and the frequency of attenuated strains in the groups with the highest numbers of strains, i.e., those in the cluster peaks in Figure 6, was further analyzed (Figure 7). A Pearson’s chi-square test of independence was performed to examine the relation between ΔG grouping and the frequency of viral attenuation or vaccine derivation/relatedness. The relation between these variables was significant, with group 36 with the higher ΔG including ~54% of attenuated/vaccine-derived/vaccine revertant strains in contrast to ~1.5% in group 1 with the lower ΔG. The observation that a higher ΔG is present in half of the attenuated/vaccine-derived/vaccine revertant strains underscores the many variations unrelated to the *NSP16* duplex that may underlie attenuation. It is hard to estimate IBV genetic diversity and mutation rates accurately, because attenuated vaccine strains can evolve in various ways and counteract an attenuation event without altering its underlying mechanism with another that causes reversal [54].

#### 2.2.3. Sequences Similar to the IBV *NSP16* Extended Duplex Are Present in the Rousettus Bat Betacoronaviruses (Nobecoviruses), Related to SARS-CoV-1/-2

Using the BLAST program, we checked for sequences similar to the IBV *NSP16* extended duplex in *Viridae* and detected them only in *Rousettus* bat coronaviruses (genus *Nobecoviruses*) (Figure 8), which are *Betacoronaviruses* related to SARS-CoV-1 and -2 and able to utilize the human ACE2 receptor for cell entry in vitro [85,86].

### 2.3. Severe Acute Respiratory Syndrome Coronavirus-2 (SARS-CoV-2) and Other Betacoronaviruses Infecting Humans

We then searched for sequences reading the same in the sense and antisense directions, with complementary halves that could form a duplex in SARS-CoV-2, a *Betacoronavirus* (subgenus *Sarbecovirus*) and the most recent coronavirus infecting humans that has been characterized. SARS-CoV-2 has a sequence with said characteristics in the region of the replicase gene encoding NSP3 (Figure 9A). The duplex-forming sequence is also part of an extended discontinuous duplex-forming region. SARS-CoV-1 has a similar duplex-forming region but as part of an extended continuous duplex (Figure 9B). These SARS-CoV regions share a secondary structural similarity, and a 10-nucleotide region of primary sequence similarity repeated twice, with the IBV extended duplex (Figure 9C). These primary and secondary structural similarities underscore the nucleotide sequence, over the encoded amino sequence of these regions, as functionally relevant to potential enhancer function and pathogenicity.

We then searched the SARS-CoV-2 genome for sequences proximal or distal to the core duplex-forming region and found several, including one immediately after TRS-L reminiscent of the MHV enhancer and others in *NSP4*, *NSP10*, and *S* (Figure 10A–C). As shown in Figure 10C, the first complementary half of the SARS-CoV-2 *NSP3* duplex-forming sequence shows similarities to the enhancer element proximal to the *N* gene in TGEV (Figure 1).

The SARS-CoV-2 NSP3 duplex, when not pairing with the 5′-UTR complementary sequence, can pair with a sequence in the accessory *ORF3a* gene with a minimum free energy similar to that of the within-duplex pairing (Figure 11A). The latter interaction may affect the expression of the envelope gene, encoding a viroporin [87] distal to *ORF3a* and *ORF3b*. Figure 11B depicts a model by which the *NSP3* duplex could open and interact with the complementary sequence in *ORF3a*, as it would with those in the other genomic locations shown in Figure 10 and Figure 11C. Interactions with other potential viral or host nucleic acids and proteins could influence the closing and opening of the duplex.

No mutations in the SARS-CoV-2 *NSP3* extended duplex were detected in a GISAID database (GISAID—Initiative) [88,89,90] search of all SARS-CoV-2 lineages, including Alpha (B.1.1.7 [Pango v.4.3.1 consensus call], Alpha [B.1.1.7-like] Scorpio, former VOC Alpha GRY [B.1.1.7 + Q. *]), Beta (B.1.351 + B.1.351.2 + B.1.351.3, former VOC Beta GH/501Y.V2), Gamma (P.1 + P.1. *, former VOC Gamma GR/501Y.V3), Delta (B.1.617.2, former VOC Delta GK [B.1.617.2 + AY. *]), Lambda (C.37 + C.37.1, former VOC Lambda GR/452Q.V1), Mu (B.1.621 + B.1.621.1, former VOI MuGH), Omicron (B.1.529 + BA. *, VOI GRA [JN.1 + JN.1. *]), BA.286 + BA.286. *, XBB.1.5 + XBB.1.5. *, XBB.1.16 + XBB.1.16. *, EG.5 + EG.5. *, BA.2.75 + BA.275. *, CH.1.1 + CH.1.1. *, XBB + XBB. *, XBB.1.9.1 + XBB.1.9.1. *, XBB.1.9.2 + XBB.1.9.2. *, XBB.2.3 + XBB.2.3. *, GH/490R (B.1.640 + B.1.640. *). The same applies to VOIs GRA XBB.1.5 + XBB.1.5. *, XBB.1.16 + XBB.1.16. *, EG.5 + EG.5. *, BA.2.86 + BA.2.86. * [excluding JN.1, JN.1. *], JN.1 + JN.1. *; and for VUMs GRA BA.2.75+ BA.2.75. *, CH.1.1 + CH.1.1. *, XBB + XBB. * [excluding XBB1.5, XBB1.16, XBB1.9.1], XBB 1.9.2 and XBB.2.3, XBB.1.9.1 + XBB.1.9.1. *, XBB.1.9.2 + XBB1.9.2. *, and XBB.2.3 + XBB.2.3. *.

The SARS-CoV-2 *NSP3* duplex-forming sequence almost identical in the sense and antisense directions and with complementary halves is present in SARS-CoV-1 (another *Sarbecovirus*) with minor differences (Figure 5). In comparison to all *Viridae*, the 36-nucleotide *NSP3* sequence in SARS-CoV-2 is present only in the *NSP3*s of related bat *Sarbecoviruses* (Figure 12A). The order of levels of similarity (expect, 1 × 10^−9^ to 3 × 10^−6^) matches that of the minimum free energy (ΔG) of the duplex structure, with more favorable structures having a lower ΔG. Sequences in SARS-CoV-2-related bat coronaviruses could be divided into four groups, as shown in Figure 12. The SARS-CoV-1 *NSP3* duplex is conserved among all SARS-CoV-1 strains in GenBank and related non-human coronaviruses, with a few exceptions that mostly do not affect the duplex structure or render it more robust.

The extended *NSP3* duplex in SARS-CoV-1 could pair with a similar sequence (e = 0.15) in *ORF3b*, instead of *ORF3a*, as for SARS-CoV-2 (Figure 12B).

SARS-CoV-1, including the Tor2 (GenBank accession: NC_004718.3) and Urbani (MT308984) strains, has another duplex-forming sequence reading the same in the sense and antisense directions with complementary halves in the *ORF3a* gene that could pair with a complementary sequence in *NSP4* approximating the TRS-L and TRS-B of the *ORF3b* gene. The pairing would be less stable than that between the SARS-CoV-1 *NSP3* duplex and the *ORF3b* gene, an interferon antagonist, upstream of the envelope gene [91]. The activities of enhancer elements in SARS-CoV-2 appear stronger than those in SARS-CoV-1 (Figure 12C).

Although SARS-CoV-2 does not have the duplex-forming potential core enhancer element in *ORF3a* present in SARS-CoV-1, the amino acid sequence encoded by the corresponding region is similar, with two conservative substitutions out of the three amino acid changes (Figure 12D).

In terms of other *Betacoronaviruses*, the Middle East respiratory syndrome (MERS)-CoV (subgenus *Merbecovirus*), which like SARS-CoV-1 and -2, has caused epidemics in humans, has sequences in the NSP3-, NSP5-, and NSP6-encoding regions of the replicase gene (shown in Figure 13) matching those of the proximal (end of replicase gene), intermediate, and distal (near the TRS-B preceding the *N* gene) elements, respectively, of the *Alphacoronavirus* TGEV (shown in Figure 1). However, the minimum free energy for the duplex pairing of the distal and proximal sequences in MERS-CoV (ΔG of −1.3 vs. −3.8 kcal/mol in TGEV) is higher and may not be functional. The observation that all the MERS-CoV sequences similar to the TGEV enhancer elements are within the replicase gene would further argue against their functionality in altering the expression of structural and accessory genes in the last third of the viral genome.

The MERS-CoV NSP16-encoding region of the replicase gene harbors a duplex-forming sequence reading similarly (one nucleotide difference out of 26 positions) in the sense and antisense directions, with complementary halves, and as part of an extended duplex region that has regions of similarity with the *NSP3* duplexes of SARS-CoV-2 and -1 (Figure 14).

Another *Betacoronavirus* infecting humans, the hCoV-OC43 of the same *Embecovirus* subgenus as murine hepatitis virus and bovine coronavirus, has a duplex reading the same in the sense and antisense directions as part of an extended duplex in *NSP3*, with sequence similarities to those in the *NSP3* of SARS-CoV-2 and -1 (Figure 15).

Finally, we analyzed similarities among all the duplexes described above for *Betacoronaviruses* infecting humans. The potential coronaviral duplex-forming enhancers contain repeats of hexanucleotide sequences, some with complementary halves (Figure 16), which might function as binding sites for host or viral proteins or RNAs, possibly regulating the closing and opening of the duplex sequences or bridging interactions with remote complementary sequences. Transcription factor binding drives enhancer activity and gene expression in various systems, and enhancers are usually intrinsically redundant in factors’ binding sequences in either orientation [92].

## 3. Discussion

In the present study, using the experimentally determined nonanucleotide-based enhancer elements of an *Alphacoronavirus* and a *Betacoronavirus* and experience with other viral and host genome enhanceosomes, we detected potential enhancers, which vary in primary sequence and location, in the phylogenetically ancient *Gammacoronavirus* IBV and more recent *Betacoronaviruses*, including SARS-CoV-2. The proposed enhancer model comprises an epistatic network with a core duplex-forming region molecular switch with identical sense and antisense sequences and complementary halves that could transition to an open configuration as dictated by a viral/host protein(s)/RNA(s), allowing a robust pairing with proximal or distal remote viral genome complementary sequences to enhance a specific downstream viral gene expression.

Increased host immune evasion provides a replicative advantage to the virus. The coronaviral enhancer models proposed and reviewed here provide an alternative potential mechanism for viral attenuation, a process with multiple underlying mechanisms yet to be fully characterized. Although we did not confirm the function of the predicted IBV enhancer with laboratory experiments, analyses of the sequenced IBV strains suggest the disruptive effects of variation in the *NSP16* gene duplex-forming region, remote complementary sequences in the spike gene upstream of the immune evasion-related *ORF3a*, or both, consistent with some documented cases of viral attenuation as potential novel pathogenicity determinants. Moreover, as assessed by minimum free energy, the distribution of the duplex robustness of the IBV enhancer element shows a dichotomization of virulence status. These analyses of sequenced IBV strains are limited by the fact that attenuation is multifactorial, and changes in nonstructural, structural, and accessory genes and their proteins may occur concomitantly in various combinations, whose effects remain to be studied.

Avian IBV, including chicken, turkey, pigeon, goose, and swan coronaviruses, among others [93], is one of the major causes of highly contagious respiratory diseases in domestic fowl and a severe economic threat to the poultry industry worldwide, as the second most dangerous disease after highly pathogenic influenza [42,50,93,94]. Live-attenuated vaccines against IBV have been generated by serial passage, from 51 to over 100 passages, of a virulent isolate in chicken embryonated eggs until attenuation is achieved [49,50]. The exact mechanisms of attenuation are unknown, and vaccine strains have a risk of reversion to virulence.

Several studies have aimed at characterizing the determinants of IBV strains’ attenuation. For instance, the attenuated recombinant IBV M41-R was generated from the pathogenic strain MR41-CK by two amino acid changes, namely, NSP10-Pro85Leu and NSP-14-Val393Leu, which were associated with a temperature-sensitive replication phenotype at 41 °C in vitro [95]. Likewise, for MHV, mutations underlying phenotypes characterized by the inability to synthesize viral RNA at a non-permissive temperature mapped to gene regions encoding the replicase–transcriptase nonstructural proteins 4, 5, 10, 12, 14, and 16 [96]. Accessory genes in IBV have also been associated with attenuation in natural hosts [47,48,97,98,99].

Although changes in the S glycoprotein have been linked to IBV host adaptation in some studies, leading to either increased or decreased pathogenicity [100,101], this has not been the case in others. For instance, even when the S glycoprotein ectodomain from the attenuated Beaudette laboratory strain was replaced with that from the pathogenic Mass41 strain, the strain remained nonpathogenic in chickens, indicating that the S glycoprotein is not the only determinant of IBV pathogenicity [50,94].

In another study, a chimeric IBV was created with the replicase genes 1a and 1ab from the attenuated Beaudette strain and all the structural genes from the pathogenic Mass 41 strain, including the *S* gene. This chimeric virus was not pathogenic in chickens, indicating that the replicase proteins also appear to be determinants of the IBV pathotype [95,102]. Genetic differences reported in ORF1a and S between virulent and avirulent strains of IBV also led others to suggest that the replicase proteins, in addition to S, are involved in the pathotype of the virus [103]. In the present study, a potential enhancer is the NSP16-encoding region of the replicase.

The MHV and SARS-CoV-2 enhancers also are predicted to affect viral gene expression in the host immune evasion. Although the duplex-forming sequence in SARS-CoV-2 appears thus far conserved among variants, its variability may adversely affect virulence. Based on the IBV experience, it may be early to detect variation in SARS-CoV-2. The same applies to SARS-CoV-1, where scarce cases of mutations do not adversely affect the minimum free energy of the duplex.

The features of the potential coronavirus enhancers are similar to those of enhancers overall. Enhancers, as cis-acting key scaffolds for the transient dynamic recruitment and assembly of transcription factor/coactivator clusters, integrate regulatory information encoded by the surrounding genome and the biophysical properties of trans-acting transcription pieces of machinery, such as RNA polymerase and transcriptional coregulators [104,105,106,107]. All the coronaviral duplex-forming structures described here have repeats of similar motifs in different orientations, possibly mediating binding to viral and host factors involved in potential enhancer function, consistent with the redundancy of transcription factor-interacting sequences within enhancers.

Enhancers control gene expression location, level, and timing [44,108,109]. In mammalian systems, enhancers determine spatiotemporal gene expression programs by engaging distant promoters over long genomic ranges [110,111,112,113,114]. For instance, some enhancer–promoter RNA interaction sites involve the pairwise interacting of Alu and non-Alu RNA sequences that tend to be complementary and potentially form duplexes [110].

Beyond the potential coronaviral enhancers analyzed here, we annotated duplex-forming regions reading similarly in the sense and antisense directions, with complementary halves, in MHV and bovine coronavirus, up to 83 nucleotides long (Figure 17). The latter two viruses illustrate the presence of possible networks of different potential enhancer elements with more than one long-range RNA-RNA interaction. They are reminiscent of the nested epistasis enhancer networks for robust genome regulation reported in mammalian genomes [115].

In terms of the possible origin of the duplex-forming regions reading similarly in the sense and antisense directions with complementary halves (inverted repeats) and a potential enhancer function, one could propose a template-switching mutation mechanism during RNA replication via RNA-dependent RNA polymerase. This mechanism would be similar to that postulated by Mönttinen et al. [116], in which a template-switching mutation mechanism during DNA replication by DNA-dependent DNA polymerase could generate hairpin structures via inverted repeats or inverted and direct repeats that could evolve into novel microRNAs [116]. The observation by Mönttinen et al. [116] of template-switching mutation-driven evolution at the nucleic acid level occurring without affecting the protein sequence encoded is consistent, for instance, with the observation here that although the duplex in SARS-CoV-1 *ORF3a* lacks a homolog in SARS-CoV-2, the encoded peptide sequences are similar. The same applies to variation in the potential IBV enhancer element at the nucleotide level, with the amino acid sequences encoded by the duplex region being conserved among strains.

The observation that the sense and antisense strands have the same or almost the same sequence in the duplexes described also raises the possibility that the enhancer activity may extend to gene expression by the antisense strand. As is the case for the transcriptomes of cytomegaloviruses [117], retroviruses [118,119,120], and prokaryotes [121,122], that of SARS-CoV-2 might include RNAs that are transcribed from the negative-sense genomic RNA and encode functional proteins or nucleic acids involved in RNA regulation [123]. For instance, HTLV-1 antisense strand-encoded mRNA interacts with the promoter and enhances transcription of the C-C chemokine receptor type 4 (*CCR4*) gene to support the proliferation of HTLV-1-infected cells, and HIV-1 antisense mRNA is recruited to the viral long terminal repeat and inhibits sense mRNA expression, to maintain the latency of HIV-1 infection [124].

The potential enhancer elements and models discussed here require direct experimental validation. However, delineating possible long-range RNA-RNA interactions can enrich prediction tools and the analysis of RNA folding and its potential for novel structures, which are challenges for in silico prediction tools, none of which can offer full accuracy [32,125]. Other enhancer mechanisms may be at play among coronaviruses, as illustrated by the fact that the enhancer elements experimentally defined in TGEV are not present in all *Alphacoronaviruses* [25].

The experience with IBV strains provides fertile ground for these explorations, illustrating the well-documented intricacies and limitations of ascribing viral attenuation to a particular mechanism, given the diversity and frequency of mutations and recombination events among an ever-growing number of variants. However, the characterization of all possible attenuation and enhancement mechanisms of viral pathogenesis bodes well for the rational development of preventive and therapeutic strategies for all coronaviruses. Similar to using combination therapies against retroviruses [123], the combination of attenuation mechanisms may allow the development of more effective coronaviral vaccines [126,127,128].

## 4. Materials and Methods

### 4.1. Rationale for Primary and Secondary Structure Analyses and Use of TGEV Enhancer Model

All the analyses performed include both primary and secondary structural levels, because enhancer elements involve both. Although tertiary, quaternary, and quinary structural levels are likely also involved, they are more challenging to assess at this point, because much remains to be determined about the nature and functions of viral and host factors, whether proteins, nucleic acids, or osmolytes, involved in the structures and functions of experimentally defined coronaviral enhancers or other gene expression regulatory elements.

We first extended the TGEV enhancer model to postulate a mechanistic model for the experimentally characterized MHV enhancer. Because the MHV nonanucleotide enhancer contains an octanucleotide that has identical sequences in the sense and antisense strand with complementary halves, we searched for this octanucleotide among coronaviruses, using the “Find in this sequence” feature in GenBank (National Library of Medicine) and the BLASTN program [129], and found it in IBV. As detailed in the next section, we searched for similar duplex-forming genomic sequences in other coronaviruses.

### 4.2. Detection of Duplex-Forming Genomic Sequences Reading Similarly in the Sense and Antisense Directions with Complementary Halves as Potential Core Enhancer Elements

We compared coronaviral genomic sequences against themselves using the BLASTN program (nucleotide collection [nr/nt]; expect threshold: 0.5; mismatch scores: 2, −3; gap costs: linear) [129], which evaluates sense and antisense strands, to determine the presence of genomic sequences reading the same, or almost the same, in the sense and antisense directions. These sequences have complementary halves because they are inverted repeats. The representative reference sequences (RefSeq) for each coronaviral genus were obtained from the International Committee on Taxonomy of Viruses (ICTV) Coronaviridae Study Group [130] and the GenBank and NCBI Virus databases.

### 4.3. Detection of Coronaviral Genomic Sequences That Could Pair to a Core Enhancer Element

We used the BLASTN program (nucleotide collection [nr/nt]; expect threshold: 10; mismatch scores: 2, −3; gap costs: linear) [129] to determine if the MHV enhancer nonanucleotide could pair with genomic sequences, following TGEV’s enhancer model to affect subgenomic mRNA expression. The locations of the detected genomic sequences and the boundaries of nonstructural, structural, and accessory open reading frames were determined, based on GenBank annotation and a manual inspection of multiple alignments and sequence similarities, as previously described [131]. The same approach was used with all potential core enhancer elements that, similar to the MHV enhancer, consisted of a duplex-forming sequence reading similarly in the sense and antisense directions with complementary halves.

### 4.4. Characterization of Extended Duplexes and Visualization of RNA Secondary Structures and Estimation of Minimum Free Energies of Duplexes

Sequences of up to 50 nucleotides before and after the core enhancer duplexes were analyzed for the possibility of their forming duplexes beyond the core duplex, reading the same or similarly in the sense and antisense directions. RNA secondary structures were visualized using forna, a force-directed graph layout (ViennaRNA Web services) [132]. Optimal secondary structures were also visualized using the RNAfold webserver, which was used to estimate the minimum free energy, reflecting the robustness of the pairings within the core and extended duplexes and with other genomic sequences [133,134].

### 4.5. Determination of Similarities Between Coronaviral Extended Duplexes with Viridae Sequences in GenBank and Analysis of Possible Nucleotide and Amino Acid Sequence Variations Being Consistent with Attenuation

The nucleotide sequences of the reference extended duplexes were compared against those entered under *Viridae* in the GenBank database. This allowed us to detect similarities between the IBV or SARS-CoV-2 duplex-forming potential core enhancer elements and sequences in bat coronaviruses related to SARS-coronaviruses. It also allowed the detection of primary sequence variations in the extended duplex among IBV strains in the GenBank database. Appendix A includes the accession numbers, collection site, and date of the IBV strains obtained from GenBank and the references therein.

The nucleotide sequences of the extended duplex of the IBV variants were translated using the insilico (DNA to protein translation (ehu.es)) [135] and Expasy (ExPASy—Translate tool [136]) tools to determine if amino acid sequence variation was consistent with viral attenuation.

The GISAID database (GISAID Initiative) [88,89,90] was used to search for nucleotide variation in the *NSP3* SARS-CoV-2 extended duplex region of all SARS-CoV-2 lineages. Each variant of concern or interest listed under ‘variants’ in the GISAID database and specified in the Section 2 was checked for mutations in the *NSP3* SARS-CoV-2 extended duplex region.

### 4.6. Assessment of Sequence Similarities in Multiple Alignments of Potential Coronaviral Extended-Duplex Core Enhancer Elements

Multiple potential coronaviral extended-duplex core enhancer elements alignments were evaluated using CLUSTAL Omega (https://www.ebi.ac.uk/services accessed on multiple occasions between January and June 2024) [137] to search for similarities, including short repeats in different orientations, which were confirmed by visual inspection.

### 4.7. Assessment of Attenuation Status of IBV Strains

To analyze the relationship between IBV duplex variation and viral attenuation, we checked publications listed in GenBank for the nucleotide sequences included and publications listing attenuated vaccine strains or strains derived from them (Appendix A). When no information could be obtained on their attenuation status, strains were categorized as non-attenuated. The analysis here focused on the strains in groups 1 and 36 in Figure 4, which were the most representative in terms of the number of entries of the two clusters, apparent by the ΔG distribution of the *NSP16* duplexes (as shown in Figure 5). Using the chi-square test, we also compared the frequencies of the attenuated and non-attenuated strains in groups 1 and 36, representing the peaks of the dichotomized clusters.

## Figures and Tables

**Figure 1 ijms-25-08012-f001:**
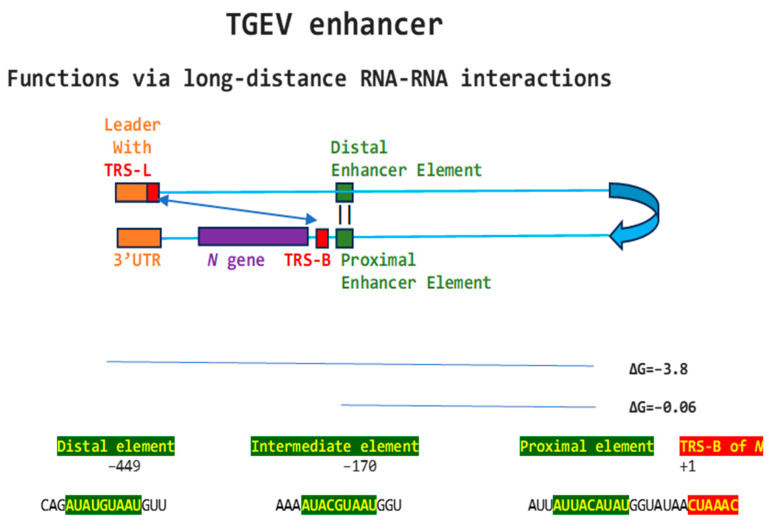
Nonanucleotide enhancer in transmissible gastroenteritis virus (TGEV, *Alphacoronavirus*, subgenus *Tegacovirus*, GenBank Accession: NC_038861). The TGEV enhancer upregulates the expression of the subgenomic RNA encoding the nucleocapsid (N) protein, possibly by approximating the TRS-L and TRS-B of N via duplex formation between the distal (close to the middle of the membrane [*M*] gene) and proximal (7 nucleotides upstream of the N TRS-B) enhancer elements. Despite its sequence similarity to the distal element, the intermediate element shown does not contribute to enhancer activity consistently with its high minimum free energy.

**Figure 2 ijms-25-08012-f002:**
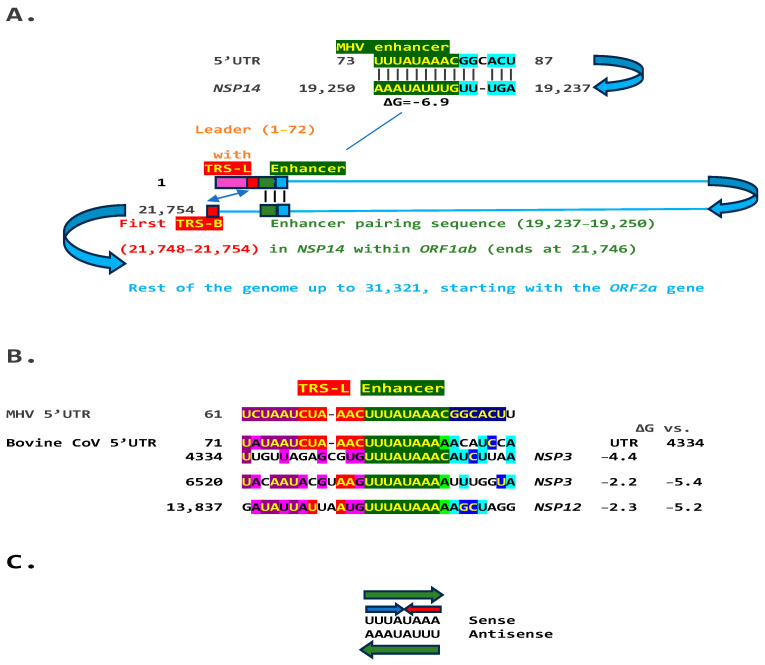
(**A**). TGEV enhancer-based model for mechanism underlying MHV (*Betacoronavirus*, subgenus *Embecovirus*, GenBank Accession: NC_048217.1 GenBank Accession: NC_048217.1) enhancer function. Located immediately after the leader sequence, the MHV enhancer (green) could pair with a sequence towards the end of the genomic region encoding the ORF1b polyprotein. Because *ORF1ab* covers approximately two-thirds of the genome, the pairing would bring the TRS-L (red) closer to the first genomic TRS-B upstream of ORF2, potentially enhancing the transcription of its subgenomic RNA in a sequence-specific manner. (**B**). Bovine CoV (*Betacoronavirus*, subgenus *Embecovirus*) has an MHV-like enhancer also immediately after the 5′-UTR leader sequence and similar sequences at three distal positions in the genome, which could pair with the 5′-UTR sequence. The minimum free energies (kcal/mol) are shown relative to the 5′-UTR sequence and the one located at position 4434; the latter pairings are predicted to be more stable. Similarities with the leader, TRS-L, enhancer, and beyond are highlighted in purple, red, green, and blue, respectively (**C**). The enhancer sequence shared between MHV and bovine CoV includes an octanucleotide reading the same in the sense and antisense strands (green arrows) and with complementary halves (blue and red arrows).

**Figure 3 ijms-25-08012-f003:**
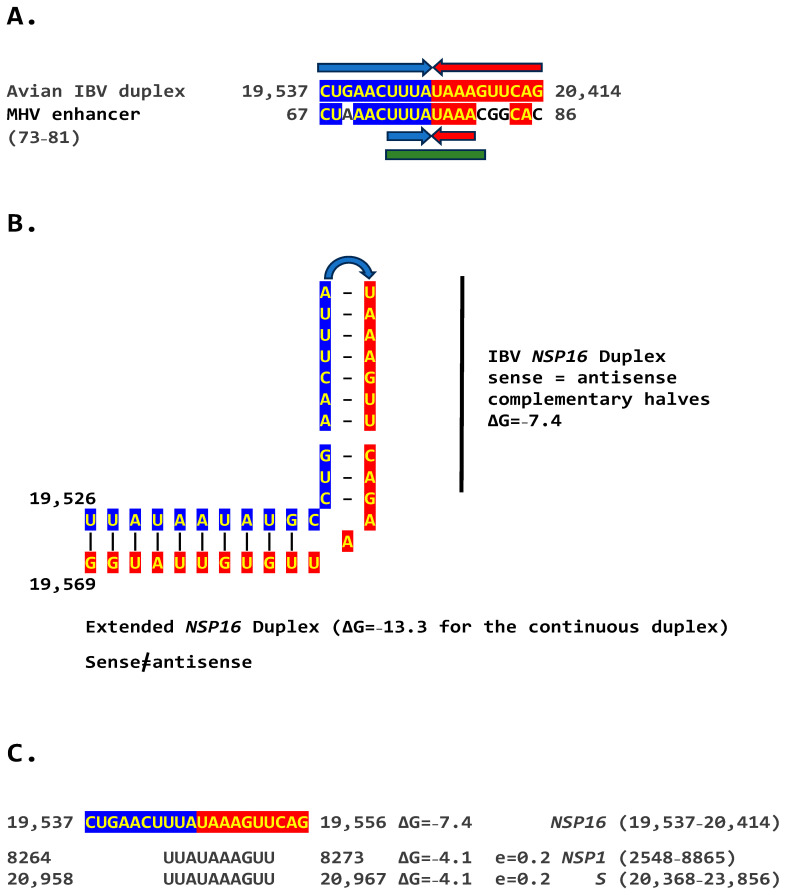
(**A**). Similarity between the *NSP16* duplex in avian infectious bronchitis virus (IBV; Gammacoronavirus, subgenus *Igacovirus*) and the MHV enhancer (green). Red and blue arrows indicate complementary halves that can form a duplex (**B**). NSP16 duplex (with same sense and antisense sequences) and extended duplex. ΔG is the minimum free energy in kcal/mol. (**C**). Similar distal sequences potentially pairing with the *NSP16* duplex. One sequence is proximal (region encoding the first protein in ORF1a) and the other distal (region encoding spike [S]) to the duplex.

**Figure 4 ijms-25-08012-f004:**
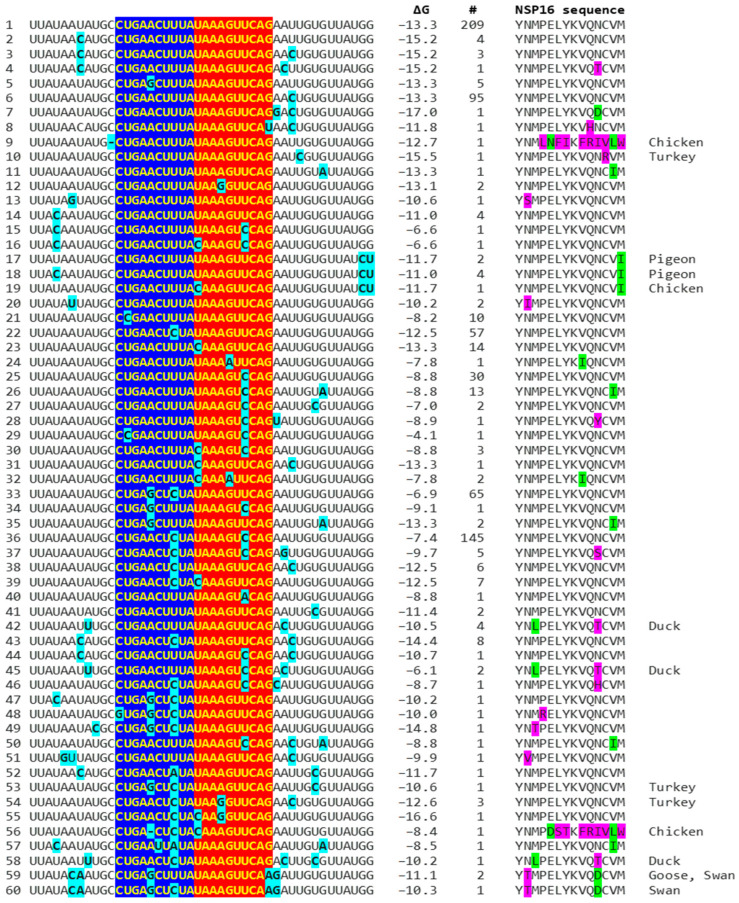
Mutations (highlighted in light blue) in avian IBV *NSP16* extended duplex (complementary halves in blue and red), duplex minimum free energy (ΔG), number of GenBank strains with each mutation combination (mutations in light blue), translated amino sequence (conservative substitutions in green, nonconservative ones in magenta), and avian IBV origin other than chicken.

**Figure 5 ijms-25-08012-f005:**
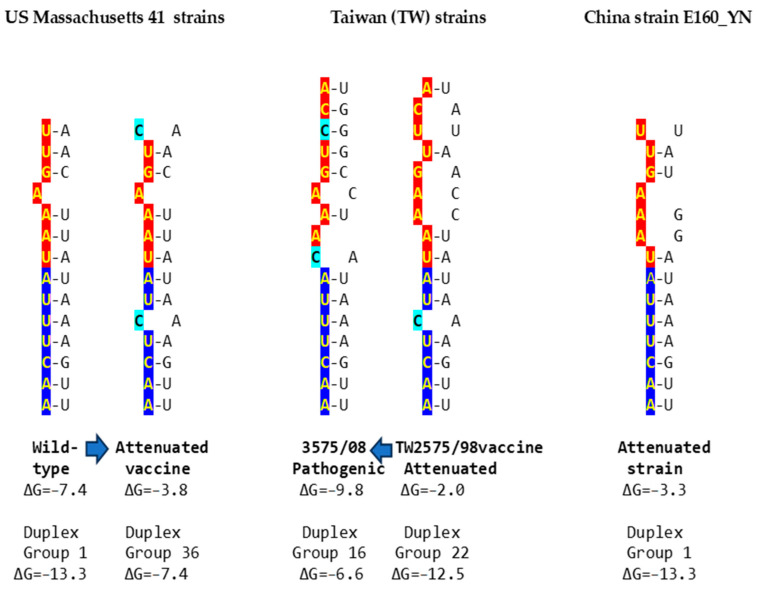
Examples of changes in the *NSP16* duplex, the distal binding sequence, or both, which may underlie published cases of viral attenuation (Massachusetts and China strains) or its reversal (Taiwan strains). The *NSP16* duplex is shown vertically in an open configuration, with complementary halves in blue and red. Mutations are highlighted in light blue.

**Figure 6 ijms-25-08012-f006:**
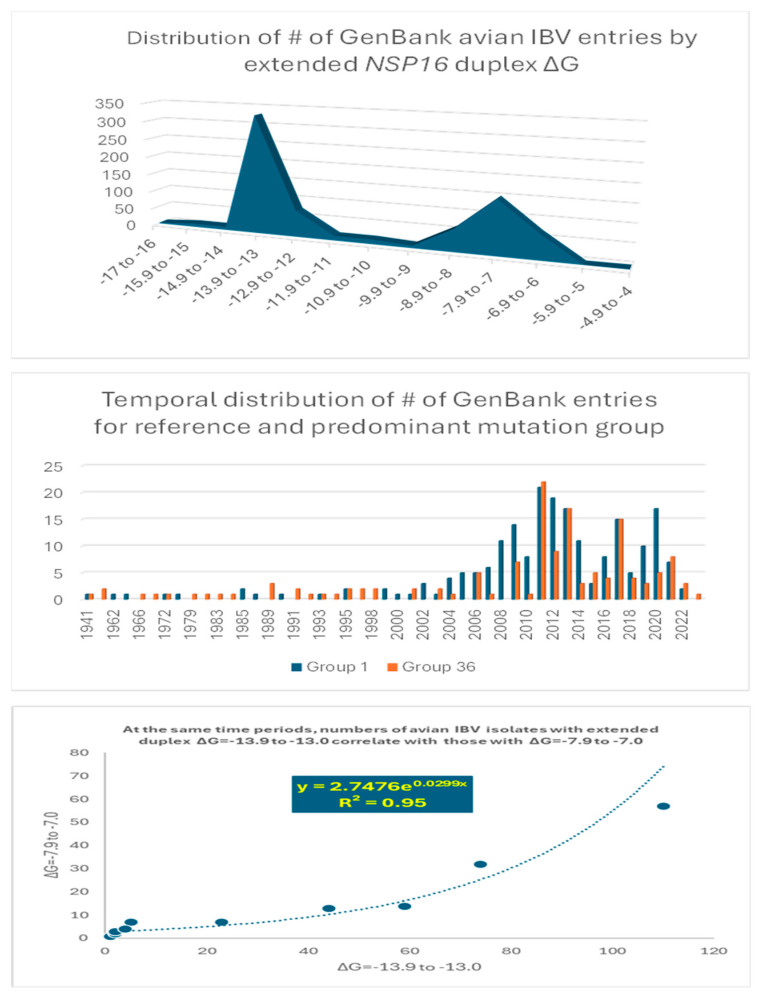
Analyses of possible association between extended *NSP16* duplex variation and viral attenuation or its reversal.

**Figure 7 ijms-25-08012-f007:**
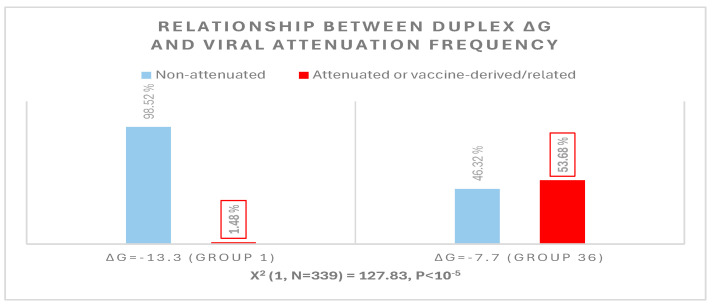
Relationship between extended *NSP16* duplex minimum free energy and frequency of viral attenuated/vaccine-derived/vaccine revertant strains.

**Figure 8 ijms-25-08012-f008:**
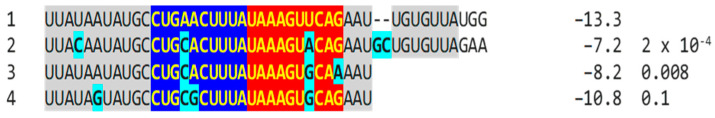
Similarities of the IBV *NSP16* extended duplex with *Rousettus* bat betacoronaviruses (*Nobecoviruses*), closely related to SARS-CoV-1 and -2, which can utilize the human ACE2 receptor in vitro. 1. Extended duplex in IBV (NC_001451). 2. OQ175246.1 (Bat CoV RlYN17 [*Rousettus leschenaultia*], China/Yunnan, 2016, isolate BtR1-BetaCoV/YN2016-Q319, toward end of ORF1ab); OQ175248.1 (Bat CoV RlYN17 [*Rousettus leschenaultia*], China/Yunnan, 2016, isolate BtR1-BetaCoV/YN2016-Q320, toward end of ORF1ab); OQ175341.1 (Bat CoV RlYN17 [*Rousettus leschenaultia*], China/Yunnan, 2017, isolate BtR1-BetaCoV/YN2017-Q321, toward end of ORF1ab starting at position 20,136). 3. MK492263.1 (*Rousettus* Bat CoV strain BtCoV92, *Cynopterus brachyotis*, Singapore, 2015). 4. OM219649.1 (Bat CoV GCCDC1, *Eonycteris spelaea*, Cambodia, 12/18,19/2010, isolate RK091); KU762332.1 (*Rousettus leschneaulti* Bat CoV isolate GCCDC1 356, China, 05/28/2014); NC_030886.1 (*Rousettus leschneaulti* Bat CoV isolate GCCDC1 356, China, 05/28/2014); KU762337.1 (*Rousettus leschneaulti* Bat CoV isolate GCCDC1 346, China, 05/28/2014); MT350598.1 (*Rousettus* bat CoV GCCDC1, *Eonycteris spelaea*, Singapore, 10/2016, betaCoV, *Nobecovirus*); OQ175331.1 (Bat CoV EsYN16, *Eonycteris spelaea*, China/Yunnan, 2016, BtEs-13BetaCoV/YN2016-Q311); OQ175332.1 (Bat CoV EsYN17, *Eonycteris spelaea*, China/Yunnan, 2017, BtEs-13BetaCoV/YN2017-Q312); OQ175333.1 (Bat CoV EsYN17, *Eonycteris spelaea*, China/Yunnan, 2017, BtEs-13BetaCoV/YN2017-Q313); OQ175242.1 (Bat CoV EsYN17, *Eonycteris spelaea*, China/Yunnan, 2017, BtEs-13BetaCoV/YN2017-Q309). Duplex complementary halves are highlighted in blue and red. Differences in bat sequences relative to IBV extended duplex are highlighted in light blue. Nucleotides shared among sequences are highlighted in gray. Minimum free energy (ΔG) is shown for each duplex, as is degree of similarity of duplexes relative to IBV reference duplex expressed as expect (e).

**Figure 9 ijms-25-08012-f009:**
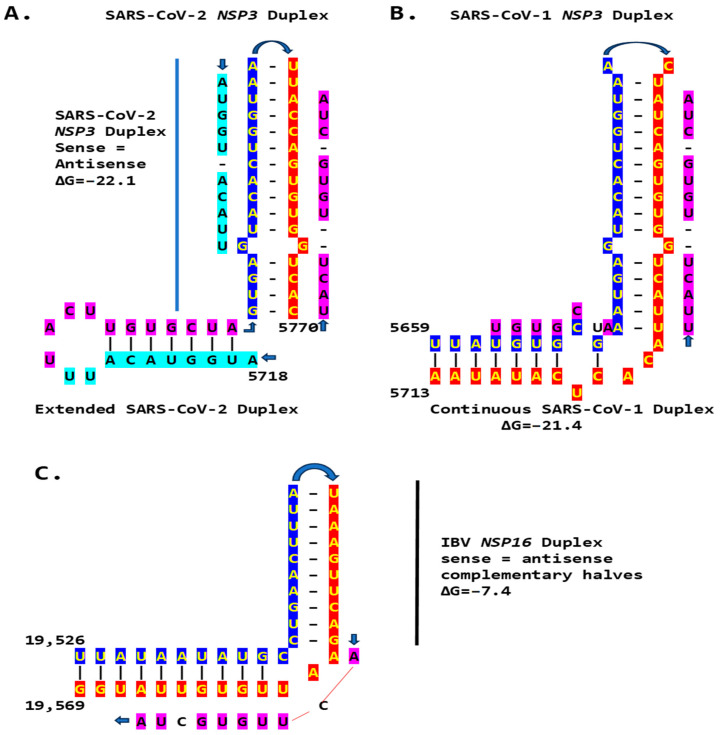
Duplex-forming sequences reading the same in the sense and antisense directions in the *NSP3* gene of SARS-CoV-2 (**A**) and SARS-CoV-1 (**B**) and in the *NSP16* of IBV (**C**). The complementary halves are highlighted in blue and red. Extended duplex regions are also shown. Regions of similarity within the SARS-CoV-2 extended duplex are highlighted in light blue, with arrows indicating that they are in inverted orientations. Similar sequences within and among SARS-CoV-2 and SARS-CoV-1 and in IBV are highlighted in pink. Minimum free energy is shown for all duplexes.

**Figure 10 ijms-25-08012-f010:**
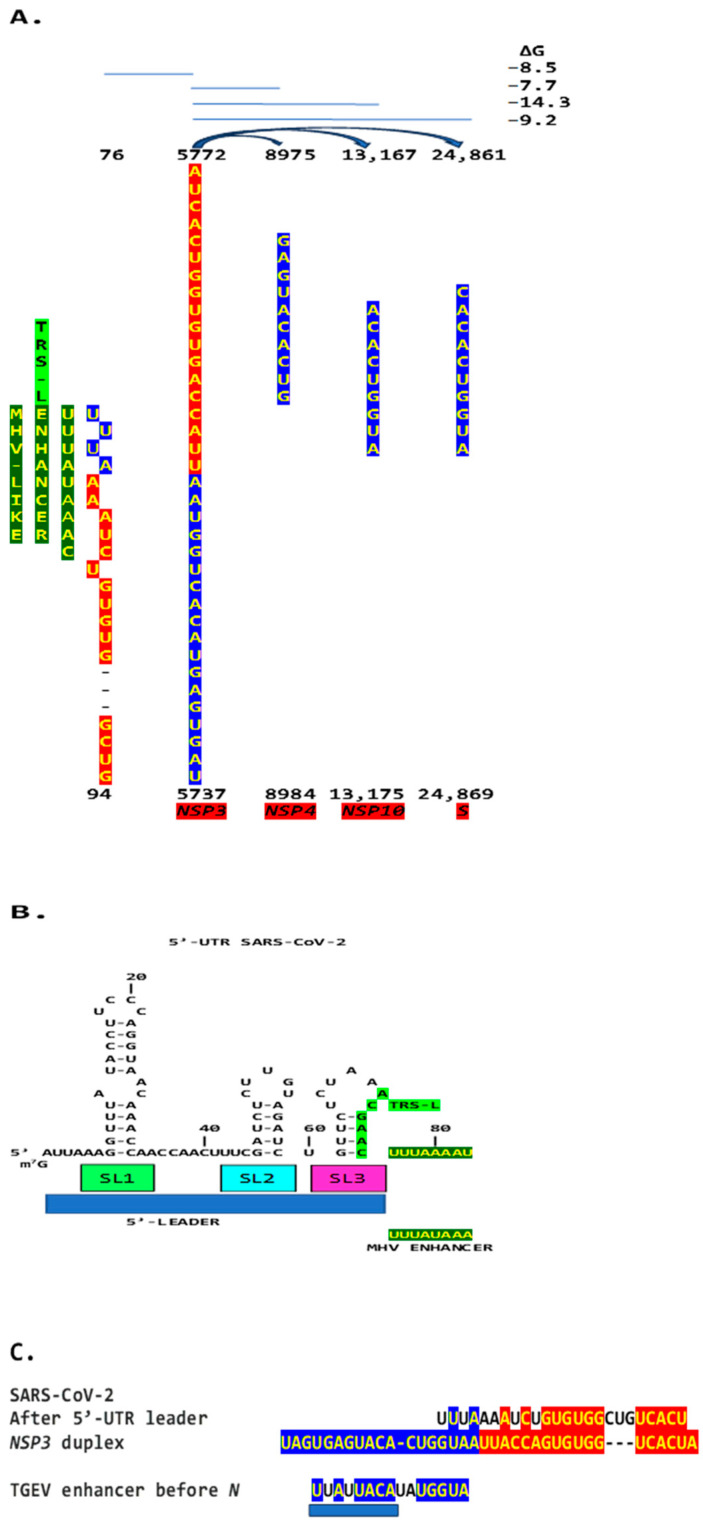
(**A**). Region in the SARS-CoV-2 5′-UTR with an MHV-like enhancer and similarity to the *NSP3* duplex, with which it can pair, leaving the duplex free to pair with three other distal genomic regions. The pairings involving the *S* gene (the first gene after *ORF1ab*; e = 2.4 for sequence similarity to the *NSP3* duplex) would approximate the TRS-L to the TRS-B of the gene encoding the viroporin *ORF3a*. The pairings involving *NSP4* and *NSP10* (e = 0.6 for both sequence comparisons to the *NSP3* duplex) would also decrease the distance between the TRS sequences. (**B**). Sequence of the SARS-CoV-2 leader (blue box) that precedes the MHV-like enhancer (yellow letters with a dark green background) and that is added via discontinuous transcription to all accessory and structural genes. (**C**). Similarities among SARS-CoV-2 5′-UTR sequence after leader, NSP3 duplex, and TGEV proximal enhancer element.

**Figure 11 ijms-25-08012-f011:**
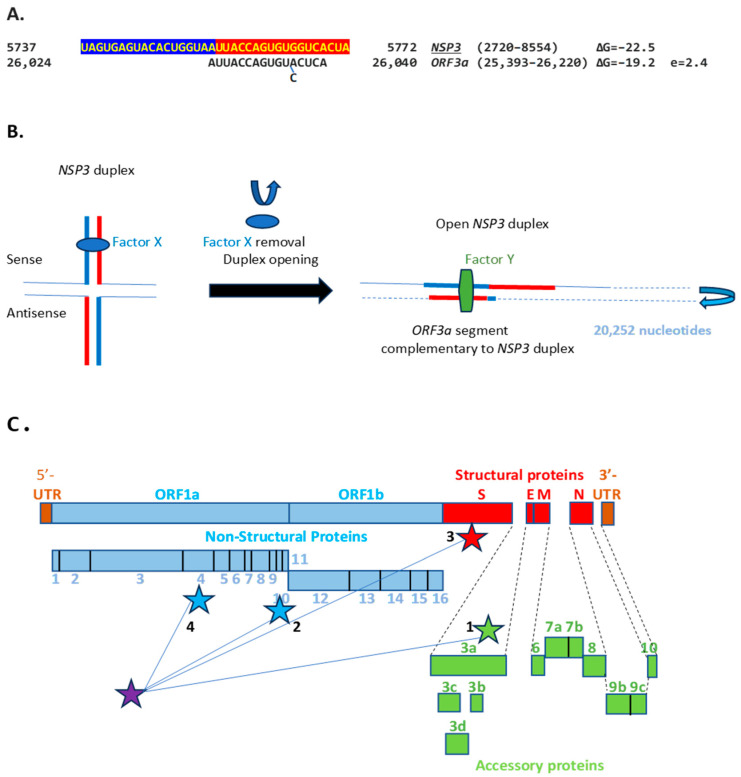
(**A**). *NSP3* duplex and complementary sequence in the SARS-CoV-2 genome. The pair can form a duplex with a minimum free energy (ΔG) similar to the *NSP3* duplex. (**B**). Switch model for the opening of the *NSP3* duplex and interaction with a 17-nucleotide complementary sequence in *ORF3a*. Initially, the duplex could be stabilized by RNA-RNA, RNA–protein, and protein–protein interactions involving undetermined factors, here termed X. Upon removal of said factors by epigenetic modification or interaction with other proteins or regulatory RNAs, the duplex could open and interact with other genomic sequences. In the illustration, the duplex and complementary sequences are 20,252 nucleotides apart, and their interaction is more stable in terms of ΔG and the length of the sequences involved than those between TRS-L (6 nucleotides long) and TRS-Bs during the discontinuous transcription of accessory and structural genes. A protein (named here Y or a combination of factors) could stabilize the new duplex between distant complementary sequences. (**C**). Positions (depicted with stars) in the SARS-CoV-2 genome of the *NSP3* duplex (purple) and complementary sequences in Panel A and Figure 10. Numbers next to positions correspond from lowest to highest ΔG; e ranged from 0.6 to 2.4.

**Figure 12 ijms-25-08012-f012:**
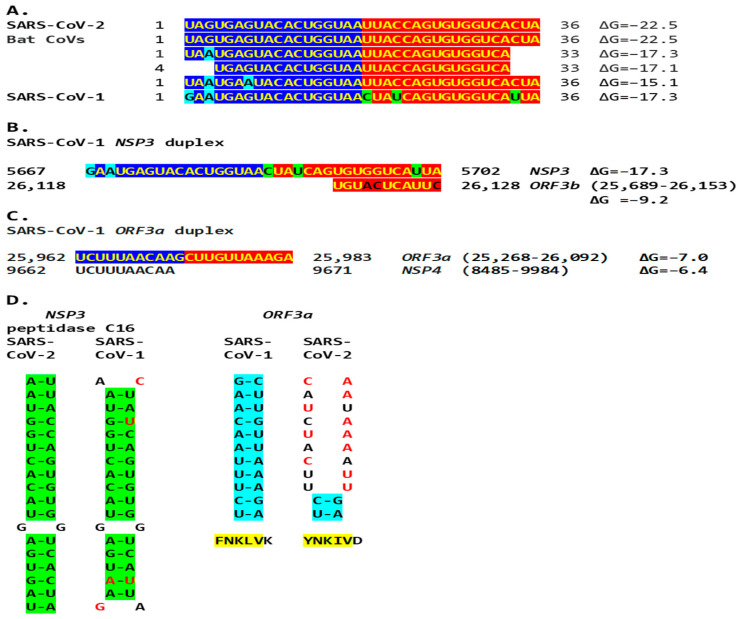
(**A**). The SARS-CoV-2 *NSP3* duplex-forming 36-nucleotide sequence is present only in closely related bat *Sarbecoviruses* and SARS-CoV-1 among all *Viridae*. Nucleotide changes among all isolates are highlighted in light blue and those unique to SARS-CoV-1 are in light green. Sequences in SARS-CoV-2-related bat coronaviruses could be divided into four groups: With an identical 36–nucleotide segment (ΔG = −22.5): BetaCoV_Yunnan_Rp_JCC9_2020 (OK287355.1); BANAL-20-236/Laos/2020 (MZ937003.2); BANAL-20-247/Laos/2020 (MZ937004.1); BANAL-20-116/Laos/2020 (MZ937002.1); BANAL-20-103/Laos/2020 (MZ937001.1); BANAL-20-52/Laos/2020 (MZ937000.1); RpYN06 strain bat/Yunnan/RpYN06/2020 (MZ081381.1); isolate PrC31 (MW703458.1). With identical nucleotides 1–33 except nucleotide 3 (ΔG = −17.3): isolates RsHB20 BtRs-BetaCoV/HB2020-Q329 (OQ175349.1), Jingmen *Rhinolophus sinicus betacoronavirus* 1 (MZ328294.1), SC2018B (OK017846.1), and BM48-31/BGR/2008 (NC_014470.1). With identical nucleotides 4–33 (ΔG = −17.1): Horseshoe bat *Sarbecovirus* isolates Rt22QT53 (OR233321.1), Rt22QT48 (OR233320.1), Rt22QT46 (OR233319.1), Rt22QT36 (OR233318.1), Rt22QT178 (OR233317.1), Rt22QT161 (OR233316.1), Rt22QT124 (OR233300.1), Rt22QB8 (OR233299.1), and Rt22QB78 (OR233298.1); and isolates BtSY1 (OP963575.1), HN2021F (OK017835.1), and HN2021E (OK017834.1). With identical nucleotides 1–36 except nucleotides 3 and 7 (ΔG = −15.1): isolates GD2019E (OK017828.1), GD2019D (OK017827.1), GD2019B (OK017826.1), GD2019A (OK017825.1), GD2017W (OK017824.1), GD2017P (OK017822.1). SARS-CoV-1 sequence in Tor2 (NC_004718) and Urbani (MT308984) strains. (**B**). A complementary segment (11 nucleotides, e = 0.3) to the SARS-CoV-1 Tor2 (NC_004718.3) and Urbani (MT308984) strains *NSP3* duplex is similar to that in SARS-CoV-2. However, the minimum free energy is higher for the pairing between the switch duplex and the complementary sequence, rendering the pairing less stable. (**C**). SARS-CoV-1 ORF3a duplex and complementary sequence in SARS-CoV-1 genome with similar minimum free energy. In this case, the duplex switch structure is distal to the complementary sequence with which it can pair with a similar ΔG. However, the effect of reducing the distance between TRS-L and the TRS-B of the gene distal to the viroporin *ORF3a*, namely the viroporin *E*, is achieved. (**D**). Comparison between the SARS-CoV-2 *NSP3* duplex (highlighted in green) and the equivalent region in SARS-CoV-1, and between the SARS-CoV-1 *ORF3a* duplex (highlighted in blue) and the equivalent region in SARS-CoV-2. The SARS-CoV-2 *NSP3* duplex is relatively well conserved in SARS-CoV-1, while the SARS-CoV-1 *ORF3a* duplex is not conserved in SARS-CoV-2 (nucleotides differing between SARS-CoV-2 and -1 are shown in red). However, the encoded amino acid sequences (highlighted in yellow) are relatively well conserved..

**Figure 13 ijms-25-08012-f013:**
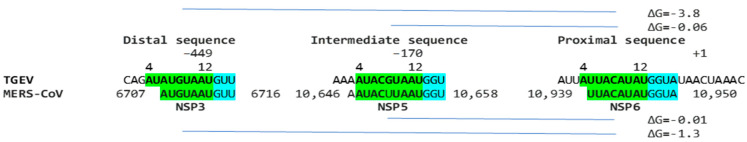
Similarity between the enhancer elements of the TGEV and MERS-CoV sequences. The intermediate sequence is the one with the highest similarity, yet in TGEV, it does not contribute to the enhancer activity. The TGEV enhancer distal and proximal sequences are partly present. The minimum free energy of the distal–proximal element pairing is higher than that in TGEV, rendering this potential enhancer unlikely to be active in MERS-CoV.

**Figure 14 ijms-25-08012-f014:**
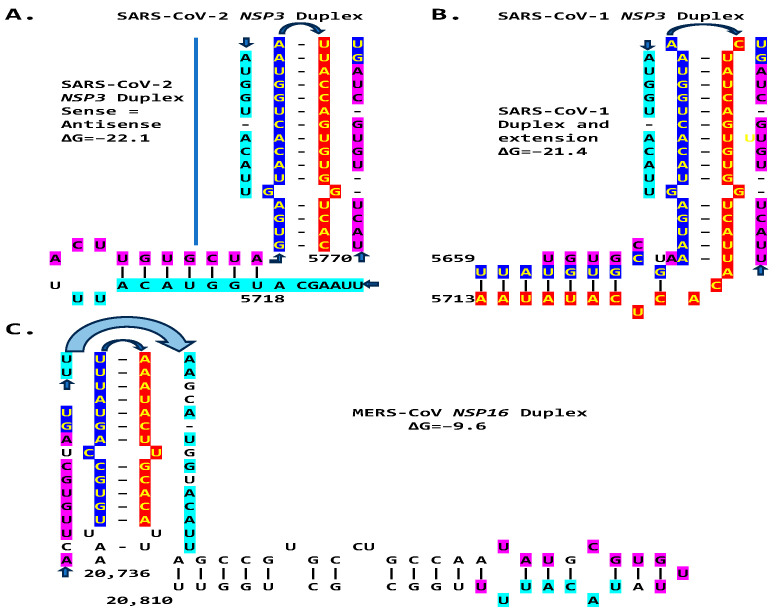
Comparison between the *NSP3* duplexes and extended duplexes of the *Betacoronaviruses* (subgenus *Sarbecovirus*) SARS-CoV-2 (**A**) and SARS-CoV-1 (**B**) with the *NSP16* duplex and extended duplex in the *Betacorinavirus* (subgenus *Merbecovirus*) MERS-CoV (**C**). Regions of similarity are highlighted in light blue and pink, and arms of the duplex reading similarly in the sense and antisense directions are highlighted in dark blue and red.

**Figure 15 ijms-25-08012-f015:**
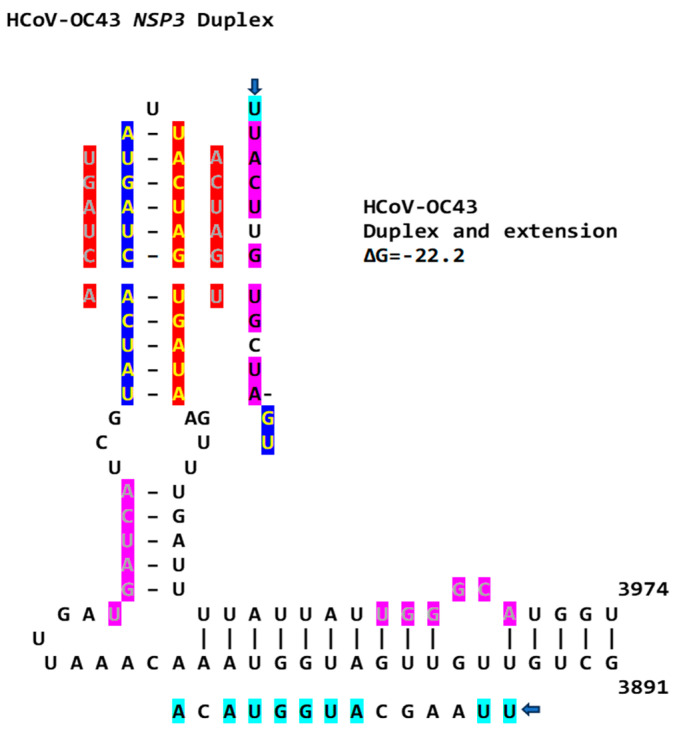
*NSP3* extended duplex in hCoV-OC43 (*Betacoronavirus*, subgenus *Embecovirus*). Regions of similarity with the NSP3 duplexes in SARS-CoV-2 and -1 and MERS-CoV are highlighted in light blue and pink, and arms of the duplex reading the same in the sense and antisense directions are highlighted in dark blue and red. Sequences in green letters highlighted in red correspond to repeated sequences in the duplexes described in this paper and shown in Figure 16.

**Figure 16 ijms-25-08012-f016:**
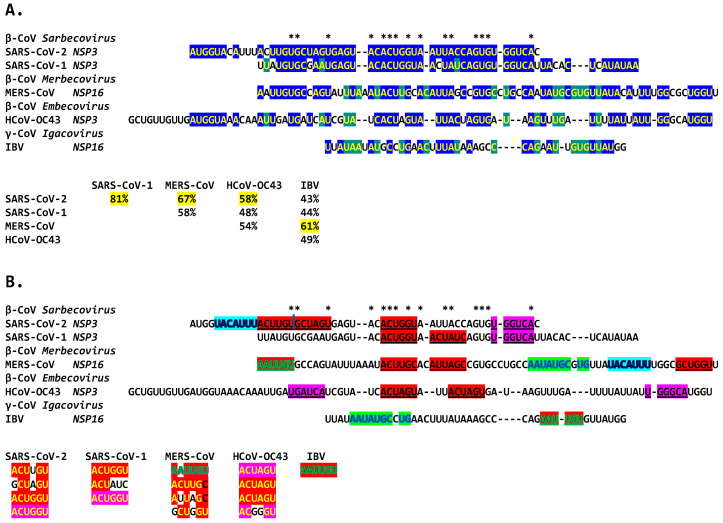
Extended duplexes of the Betacoronaviruses infecting humans. (**A**). Multiple alignments of the duplexes with similarity percentages. The highest duplex similarity was between SARS-CoV-2 and -1, while the MERS-CoV extended duplex sequence was closest to that of IBV (61% similarity), both in *NSP16* but also to the *NSP3* duplex of SARS-CoV-2 and -1 (67% and 58%). Asterisks denote nucleotide positions conserved among all duplexes, highlighted in either blue or green. (**B**). Repeated sequences within duplexes and similarity among them. Repeated similar sequences within and among duplexes are highlighted in red, fuchsia (reverse of those in red), light blue (shared between SARS-CoV-2 and MERS-CoV), and green (shared between MERS-CoV and IBV).

**Figure 17 ijms-25-08012-f017:**
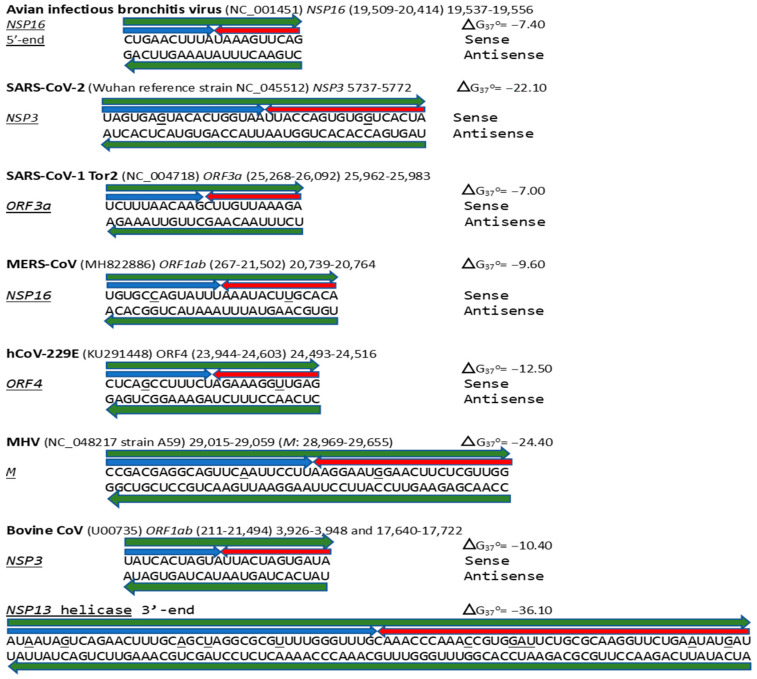
Annotated coronaviral duplex-forming sequences that read similarly in the sense and antisense directions, with complementary halves. Minimum free energies (ΔG) are shown next to each hairpin. GenBank accession numbers are shown in parentheses. Positions that do not read the same in the sense and antisense directions are underlined. The MERS-CoV *NSP16* duplex is present in bat *Merbecoviruses* and pangolin CoV HKU4 (GenBank OM009282.1). Most of the MHV *M* duplex is in the bovine coronavirus M gene (OP866729.1), and the one in the bovine CoV *NSP13* helicase is in the canine respiratory coronavirus (ON133844.1), also an *Embecovirus*. The sequences annotated here do not comprise an exhaustive list, and other coronaviruses that share some of these duplexes are mentioned in the text.

## Data Availability

All new data created and datasets analyzed are specified in the manuscript text.

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
