# Peer review of "Potential Transcriptional Enhancers in Coronaviruses: From Infectious Bronchitis Virus to SARS-CoV-2"

_ijms, 2024, doi:10.3390/ijms25158012_

Round 1

Reviewer 1 Report

Comments and Suggestions for Authors

This study detected via in silico primary and secondary structural analysis potential enhancers in various coronaviruses, from the phylogenetically ancient avian IBV to the recently emerged SARS-CoV-2. Overall, this study is relatively well-executed, and the data are consistent between experiments. However, there are still some minor issues that needs to be resolved in this manuscript.

The following are some comments and suggestions that are given to improve the manuscript:

1. It is suggested to add a list of abbreviations in the end of manuscript.

2. The author had clearly stated the research objectives and rationale in this study. Could the author briefly explain the selection foundation of coronaviruses, further verification focused only on the TGEV, not PEDV.

3. Lines 18-19: Please modify the description of the sentence “The open state would permit the enhancer to pair with remote sequences in the viral genome and modulate the expression of crucial genes involved in viral replication and host immune evasion.”.

4. While brevity is appreciated, I feel the method section is a little brief, and I am missing several technical information that would be valuable to readers.

5. The number of Figures presentation is a litter too much, It is recommended to put some figures in the supplementary materials.  

Comments on the Quality of English Language

Minor editing of English language required

Author Response

This study detected via in silico primary and secondary structural analysis potential enhancers in various coronaviruses, from the phylogenetically ancient avian IBV to the recently emerged SARS-CoV-2. Overall, this study is relatively well-executed, and the data are consistent between experiments. However, there are still some minor issues that needs to be resolved in this manuscript.

The following are some comments and suggestions that are given to improve the manuscript:

  1. It is suggested to add a list of abbreviations in the end of manuscript.

The following list has been added after the ‘Materials and Methods’ section:

Abbreviations: ACE2, angiotensin-converting enzyme 2; CCR4, C-C chemokine receptor type 4; CoV, coronavirus; hCoV, human coronavirus; HIV, human immunodeficiency virus; HTLV, Human T-lymphotropic virus; IBV, infectious bronchitis virus; IFN, interferon; MERS-CoV, M gene: membrane glycoprotein gene; Middle East respiratory syndrome coronavirus; MHV, murine hepatitis virus; mRNA, messenger ribonucleic acid; NSP, nonstructural protein; ORF, open reading frame; S, Spike protein; SARS-CoV-2, severe acute respiratory syndrome-2; TGEV, transmissible gastroenteritis virus; TRS, transcription regulatory sequence; UTR, untranslated region.

  1. The author had clearly stated the research objectives and rationale in this study. Could the author briefly explain the selection foundation of coronaviruses, further verification focused only on the TGEV, not PEDV.

The enhancer elements experimentally determined in the Alphacoronavirus TGEV (reference 25) are present in other Alphacoronaviruses, such as Feline infectious peritonitis virus (FIPV) and canine enteric coronavirus (CECoV), but not in the most distantly related Alphacoronaviruses PEDV, and human CoV-229E, as per the same publication.

We also added to the Introduction (last sentence in first and beginning of second page): ‘This nonanucleotide TGEV enhancer is conserved among other Alphacoronaviruses, such as feline infectious peritonitis virus and canine enteric coronavirus, but not porcine epidemic diarrhea virus (PEDV) or hCoV-229E [25].

The first sentence in the Discussion now reads: ‘In the present study, using the experimentally-determined nonanucleotide-based enhancer elements of an Alphacoronavirus and a Betacoronavirus and experience with other viral and host genome enhanceosomes, we detected potential enhancers, which vary in primary sequence and location, in the phylogenetically ancient Gammacoronavirus IBV and more recent Betacoronaviruses including SARS-CoV-2.’ In this last sentence, we previously wrote Alphacoronaviruses (instead of an Alphacoronavirus), which is incorrect because not all have a similar enhancer element to that characterized for TGEV.

In the penultimate paragraph of the Discussion, we added: ‘Other enhancer mechanisms may be at play among coronaviruses, as illustrated by the fact that the enhancer elements experimentally defined in TEGV are not present in all Alphacoronaviruses (25).’

  1. Lines 18-19: Please modify the description of the sentence “The open state would permit the enhancer to pair with remote sequences in the viral genome and modulate the expression of crucial genes involved in viral replication and host immune evasion.”.

The sentence now reads: The duplex open state would pair with remote proximal or distal viral genomic sequences and modulate the expression of downstream crucial genes involved in viral replication and host immune evasion.

  1. While brevity is appreciated, I feel the method section is a little brief, and I am missing several technical information that would be valuable to readers.

We added a new subsection to the beginning of the Materials and Methods:

4.1. Rationale for primary and secondary structure analyses and use of the TGEV enhancer model

All analyses performed include both primary and secondary structural levels because enhancer elements involve both. Although tertiary, quaternary and quinary structural levels are likely also involved, they are more challenging to assess at this point because much remains to be determined about the nature and functions of viral and host factors, whether proteins, nucleic acids or osmolytes, involved in the structures and functions of experimentally defined coronaviral enhancers or other gene expression regulatory elements.

We first extended the TEGV model to postulate a mechanistic model for the experimentally characterized MHV enhancer. Because the MHV nonanucleotide enhancer contains an octanucleotide that has identical sequences in the sense and antisense strand with complementary halves, we searched for this octanucleotide among coronaviruses using the “Find in this sequence” feature in GenBank (National Libray of Medicine) and the BLASTN program [130] and found it in IBV. As detailed in the next section, we searched for similar duplex-forming genomic sequences in other coronaviruses.

We also added the following sentences to the ends of the following subsections:

Subsection 4.4:
Optimal secondary structures were also visualized using the RNAfold webserver, which was used to estimate the minimum free energy reflecting the robustness of the pairings within the core and extended duplexes and with other genomic sequences [132,133].

Subsection 4.5:

Each variant of concern or interest listed under variants and specified in the Results section was checked for mutations in the NSP3 SARS-CoV-2 extended duplex region.

Subsection 4.7:

Using the Chi-square test, we also compared the frequencies of attenuated and non-attenuated strains in groups 1 and 36.

  1. The number of Figures presentation is a litter too much, It is recommended to put some figures in the supplementary materials.  

We blended four figures (Figures 12-15) into one (Figure 12).

The revised manuscript text includes other minor edits highlighted in green. Thank you for your comments, which improved the manuscript.

Reviewer 2 Report

Comments and Suggestions for Authors

 In the present manuscript, the authors analyze through an in silico study the important central regions that by creating a duplex structure may be crucial for coronavirus replication and evasion. In this study they analyze many coronaviruses. The authors need to resolve only a few issues:

1. Resolve all abbreviations when first encountered, I list a few examples: (Line15: IBV Infectious Bronchitis virus
Line 21: ORF3a (Open Reading Frame 3a)
Line 43: N gene ( nucleocapsid)
Line 49 N protein ( ditto line 43)
Line 60: M gene ( membrane Glycoprotein M)
Line 53 MHV (mouse hepatitis virus))

2. It would be appropriate in both the introduction and discussion to also comment on the studies of Drs. Sawicki ( Sawicki SG, Sawicki DL. Coronavirus transcription: a perspective. Curr Top Microbiol Immunol. 2005;287:31-55. doi:10.1007/3-540-26765-4_2; Sawicki SG, Sawicki DL, Younker D, et al. Functional and genetic analysis of coronavirus replicase-transcriptase proteins [published correction appears in PLoS Pathog. 2006 Feb;2(2):e17]. PLoS Pathog. 2005;1(4):e39. doi:10.1371/journal.ppat.0010039)

3. Supplementary materials should be organized in tables, as presented they are difficult to understand.

Author Response

In the present manuscript, the authors analyze through an in silico study the important central regions that by creating a duplex structure may be crucial for coronavirus replication and evasion. In this study they analyze many coronaviruses. The authors need to resolve only a few issues:

  1. Resolve all abbreviations when first encountered, I list a few examples: (Line15: IBV Infectious Bronchitis virus
    Line 21: ORF3a (Open Reading Frame 3a)
    Line 43: N gene ( nucleocapsid)
    Line 49 N protein ( ditto line 43)
    Line 60: M gene ( membrane Glycoprotein M)
    Line 53 MHV (mouse hepatitis virus))

The abbreviations have been resolved when first encountered. A list of Abbreviations has also been added after the Materials and Methods section.

Abbreviations: ACE2, angiotensin-converting enzyme 2; CCR4, C-C chemokine receptor type 4; CoV, coronavirus; hCoV, human coronavirus; HIV, human immunodeficiency virus; HTLV, Human T-lymphotropic virus; IBV, infectious bronchitis virus; IFN, interferon; MERS-CoV, M gene: membrane glycoprotein gene; Middle East respiratory syndrome coronavirus; MHV, murine hepatitis virus; mRNA, messenger ribonucleic acid; NSP, nonstructural protein; ORF, open reading frame; S, Spike protein; SARS-CoV-2, severe acute respiratory syndrome-2; TGEV, transmissible gastroenteritis virus; TRS, transcription regulatory sequence; UTR, untranslated region.

  1. It would be appropriate in both the introduction and discussion to also comment on the studies of Drs. Sawicki ( Sawicki SG, Sawicki DL. Coronavirus transcription: a perspective. Curr Top Microbiol Immunol. 2005;287:31-55. doi:10.1007/3-540-26765-4_2; Sawicki SG, Sawicki DL, Younker D, et al. Functional and genetic analysis of coronavirus replicase-transcriptase proteins [published correction appears in PLoS Pathog. 2006 Feb;2(2):e17]. PLoS Pathog. 2005;1(4):e39. doi:10.1371/journal.ppat.0010039)

Thank you for pointing out these seminal papers, which we have added to the Introduction and Discussion.

The pertinent paragraph in the Discussion now reads: ‘Several studies have aimed at characterizing determinants of IBV strains' attenuation. For instance, the attenuated recombinant IBV M41-R was generated from the pathogenic strain MR41-CK by two amino acid changes, namely NSP10-Pro85Leu and NSP-14-Val393Leu, which were associated with a temperature-sensitive replication phenotype at 41oC in vitro [96]. Likewise, for MHV, mutations underlying phenotypes characterized by the inability to synthesize viral RNA at a non-permissive temperature mapped to gene regions encoding the replicase-transcriptase nonstructural proteins 4, 5, 10, 12, 14, and 16 [97]. Accessory genes in IBV have also been associated with attenuation in natural hosts [48,49,98-100].

  1. Supplementary materials should be organized in tables, as presented they are difficult to understand.

We added the following boxed note at the beginning of the description of the IBV strains in the Supplement:

This section details the animal source, geographical location, year of collection, and attenuation status when known of the IBV strains in GenBank used to analyze possible associations between mutations in the NSP16 duplex and viral attenuation status. The strain GenBank accession numbers are shown in bold and highlighted in grey. A wild-type designation is highlighted in magenta, a pathogenic phenotype is highlighted in yellow, and vaccine/vaccine-derived/attenuated strains are highlighted in red. Groups 1 and 36 represent the peaks of the two clusters when strains are organized by the minimum free energy of the NSP16 duplex. Visual inspection of these two groups shows scarce red highlights in group 1 in contrast to their preponderance in group 36.

We tried to organize the information in Tables, but they occupy more than one page, especially those for groups 1 and 36, or entail one to a handful of rows. That is why we included the histogram summaries in Figures 6 and 7 of the manuscript leaving the supplement as the data source for the figures. Figure 5 in the manuscript and Supplement 1 also shows the mutations by group and the number of strains in each. We hope that the figures in the paper are appropriate summary representations of the extensive data in the Supplement section.

The revised manuscript text includes other minor edits highlighted in green. Thank you for your comments, which improved the manuscript.

Reviewer 3 Report

Comments and Suggestions for Authors

The authors detected potential enhancers in Gammacoronavirus IBV and more recent Betacoronaviruses including SARS-CoV-2. The identified potential enhancers possess a core duplex-forming region functions as molecular switches directed by viral or host factors. The topic is interesting and important. Please see the suggestions below.

1.       Line 53 in page 2, figure 1B.  Cannot find figure1 B, looks like only Figure 1.

2.       Line 147-172 (page 5 and 6),  all these lines discuss the findings of other studies, not the finds of this study. Therefore, these lines need to move to discussion section.

3.       These enhancers were identified by models, are there any assays performed to confirm function of these enhancers?

Author Response

The authors detected potential enhancers in Gammacoronavirus IBV and more recent Betacoronaviruses including SARS-CoV-2. The identified potential enhancers possess a core duplex-forming region functions as molecular switches directed by viral or host factors. The topic is interesting and important. Please see the suggestions below.

  1. Line 53 in page 2, figure 1B.  Cannot find figure1 B, looks like only Figure 1.

Thank you for pointing this out. Figure 1B is not mentioned now.

  1. Line 147-172 (page 5 and 6),  all these lines discuss the findings of other studies, not the finds of this study. Therefore, these lines need to move to discussion section.

We accordingly moved the mentioned lines from the Results to the Discussion section (highlighted in yellow in the revised version) and renumbered the references.

  1. These enhancers were identified by models, are there any assays performed to confirm function of these enhancers?

The first sentence in the Discussion now underscores that the analysis was based on two experimentally determined enhancer elements in TEGV and MHV: ‘In the present study, using the experimentally-determined nonanucleotide-based enhancer elements of an Alphacoronavirus and a Betacoronavirus and experience with other viral and host genome enhanceosomes, we detected potential enhancers, which vary in primary sequence and location, in the phylogenetically ancient Gammacoronavirus IBV and more recent Betacoronaviruses including SARS-CoV-2.’

The IBV and the Betacoronaviruses models are based on the long-range RNA-RNA interactions-based TEGV model. Although we did not directly confirm the function of the predicted IBV enhancer with laboratory experiments, our analysis of the correlates of sequence variation in the predicted enhancer element among circulating strains of IBV showed that they coincide with documented cases of viral attenuation. As assessed by minimum free energy, the distribution of duplex robustness of the IBV enhancer element also shows dichotomization of virulence status. Throughout the manuscript, we acknowledge that the predicted enhancers are ‘potential’ and require further validation beyond the indirect one provided for IBV.

The second paragraph of the Discussion section now reads: ‘Increased host immune evasion provides a replicative advantage to the virus. The coronaviral enhancer models proposed and reviewed here provide an alternative potential mechanism for viral attenuation, a process with multiple underlying mechanisms yet to be fully characterized. Although we did not confirm the function of the predicted IBV enhancer with laboratory experiments, analyses of sequenced IBV strains suggest disruptive effects of variation in the NSP16 gene duplex-forming region, remote complementary sequences in the Spike gene upstream of the immune evasion-related ORF3a, or both, consistent with some documented cases of viral attenuation as potential novel pathogenicity determinants. Moreover, as assessed by minimum free energy, the distribution of duplex robustness of the IBV enhancer element shows dichotomization of virulence status. The analyses of sequenced IBV strains are limited by the fact that attenuation is multifactorial, and changes in nonstructural, structural and accessory genes and their proteins may occur concomitantly in various combinations whose effects remain to be studied.

The revised manuscript text includes other minor edits highlighted in green. Thank you for your comments, which improved the manuscript.

Round 2

Reviewer 3 Report

Comments and Suggestions for Authors

this is an updated version and the authors have taken suggestions into consideration.